# Three New Species of *Lactifluus* (Basidiomycota, Russulaceae) from Guizhou Province, Southwest China

**DOI:** 10.3390/jof9010122

**Published:** 2023-01-15

**Authors:** Xiu-Hong Xu, A-Min Chen, Nan Yao, Ting-Chi Wen, Yun Pei, Wan-Ping Zhang

**Affiliations:** 1College of Agriculture, Guizhou University, Guiyang 550025, China; 2Institute of Vegetable Industry Technology Research, Guizhou University, Guiyang 550025, China; 3The Mushroom Research Centre, Guizhou University, Guiyang 550025, China; 4Guiyang Vegetable Technology Extension Station, Guiyang 550025, China; 5Guiyang Rural Revitalization Service Center, Guiyang 550025, China; 6The Engineering Research Center of Southwest Bio-Pharmaceutical Resources, Ministry of Education, Guizhou University, Guiyang 550025, China

**Keywords:** new taxa, ectomycorrhizal fungi, fungal morphology, fungal phylogeny, taxonomy

## Abstract

*Lactifluus* is a distinct genus of milkcaps, well known as ectomycorrhizal fungi. The characteristics of the genus *Lactifluus* include grayish-yellow, orange to orange-brown, or reddish-brown pileus, white latex from the damaged lamellae, discoloring to a brownish color, reticulate spore ornamentation, lampropalisade-type pileipellis, and the presence of lamprocystidia. Guizhou Province is rich in wild mushroom resources due to its special geographical location and natural environment. In this study, three novel *Lactifluus* species were identified through the screening of extensive fungal resources in Suiyang County, Guizhou Province, China, sampled from host species of mostly *Castanopsis* spp. and *Pinus* spp. Based on critical morphology coupled with nuclear sequences of genes encoding large subunit rRNA, internal transcribed spacer, and RNA polymerase II, these new species, *Lactifluus taibaiensis*, *Lactifluus qinggangtangensis*, and *Lactifluus jianbaensis*, were found to belong to *Lactifluus* section *Lactifluus*. A comparison with closely related species, *Lactifluus taibaiensis* was distinguished by its lighter-colored pileus, different colors of lamellae, and more subglobose basidiospores; *Lactifluus jianbaensis* was identified by the height of the spore ornamentation and its subglobose basidiospores; and *Lactifluus qinggangtangensis* was characterized by having smaller basidiospores, ridges, and pleurolamprocystid.

## 1. Introduction

*Lactifluus* (Pers.) Roussel belongs to the Russulaceae (Russulales) and is a genus of milkcaps, which is predominantly represented in subtropical and tropical regions [1,2,3,4,5]. The genus contains approximately 389 taxa (www.indexfungorum.org, accessed 31 October 2022), although De Crop (2016) estimated that there might be up to 530 *Lactifluus* species on Earth [6]. A significant number of new *Lactifluus* species have been discovered in the past ten years [7,8,9,10]. These species are mainly distinguished by their velvety pileus and stipe and a pleurotoid milkcap [1,11,12]. The hymenophoral trama of *Lactifluus* species has spherical cells (sphaerocytes), and the pileipellis structure and hymenium frequently contain cells with thicker cell walls [7,8].

*Lactifluus* is well known for the existence of several species complexes [12,13,14]. For example, *Lactifluus volemus* was discovered to have about 45 different clades [15,16,17], whereas *Lactifluus piperatus* is estimated to contain over 30 clades [18,19]. At the species level, the similarity of DNA data of *Lactifluus* species with similar or even the same morphology is low [11,15,16,17]. Groups with large differences in morphological characteristics show close kinship [7,20], and there are many cryptic and pseudocryptic species [12,14,15,19,21]. Furthermore, with the extensive use of DNA data, the sequences of the internal transcribed spacer (ITS), the large subunit 28S rRNA region (nrLSU), and the region between conserved domains 6 and 7 of the second largest subunit of RNA polymerase II (RPB2) are often used to identify *Lactifluus* species [20,22,23,24,25,26,27].

The greatest diversity of the genus is known from the Afrotropics, with 78 described species, although *Lactifluus* is also well represented in Asia, with 58 described species [8]. The genus is found in a wide range of vegetation types, including tropical and subtropical rainforests, subtropical dry forests, monsoon forests, tree savannahs, Mediterranean woodlands, temperate broadleaf and coniferous forests, and montane forests [23,28,29,30]. Host plants for *Lactifluus* species are members of the Betulaceae, Dipterocarpaceae, and Fagaceae families [16,31,32,33,34]. *Lactifluus* are commonly found in soil [27]. In order to adapt to the environment, some *Lactifluus* species with smaller basidiocarps and pleurotoid milkcaps are discovered on the stems and epigeous roots of trees, such as *Lactifluus brunellus*, *Lf. multiceps* [23], and *Lf. raspei* [35].

As a famously ectomycorrhizal fungus, *Lactifluus* contains a large number of wild edible mushrooms that are widely consumed worldwide [8,36]. Since it has a milk-like exudate, *Lactifluus* is easy to identify, except for some species of *Lactifluus piperatus* and *Lactifluus vellereus* complex groups that exude a spicy milk, which may cause discomfort after eating [37]. The scene of selling *Lactifluus* is more common in Europe, Central America, North America, and Asia [38,39,40,41,42,43], and they have a considerable socioeconomic value. A very recent account of edible mushroom species at the global scale lists some 100 edible milkcap species [44]. In particular, the *Lactifluus* sect. *Lactifluus,* such as *Lactifluus tenuicystidiatus* (X.H. Wang and Verbeken) X.H. Wang and *Lactifluus volemus* (Fr.) Kuntze have been eaten in Guizhou and Yunnan for many years and are locally called “naijiangjun” or “red naijiangjun”, being common, local, wild, edible fungi [16,39,45,46,47,48,49,50]. Several *Lactifluus* species, including *Lf. bertillonii, Lf. rugatus, Lf. volemus*, and *Lf. vellereus*, have also been shown to contain bioactive secondary metabolites, primarily sterols, sesquiterpenes, and sugar alcohols [51,52,53,54,55,56].

Southwest China includes Sichuan, Yunnan, and Guizhou Provinces, together with Chongqing City [57]. Geographically, southwest China is divided into the southeast of the Qinghai Tibet Plateau, the Sichuan Basin, and most of the Yunnan–Guizhou Plateau. The sub-region is divided into east and west sub-regions from Yalong River in the north to Nanpan River in the south of Kunming and the Hengduanshan sub-region in the northwest. The terrain structure is mainly plateaus and mountains with complex natural environmental conditions and diverse climates [58], which make the area rich in biological resources [59,60,61,62,63]. During numerous macrofungal surveys in the coniferous forests in the Wumeng Mountains of Suiyang County, Guizhou Province, three new *Lactifluus* species were discovered, based on detailed macro- and micro-morphological observations with descriptions, color photographs, and the sequence analyses of the ITS, LSU, and RPB2 regions. The objective of this research was to provide new evidence for understanding the distribution ranges and species diversity of *Lactifluus* species in China.

## 2. Material and Methods

### 2.1. Study Site and Specimen Collection

Eight specimens of the three new species were collected from Zunyi City, Suiyang County, Guizhou Province, China. Morphological descriptions were based on detailed field notes. Color names and codes were referred to HTML color codes (http://www.htmlcolorcode.org/ accessed on 1 November, 2022) [24]. The collections were dried with an electrical dryer at 50~60 °C until fully dry. Voucher specimens were deposited in the Cryptogamic Herbarium, Kunming Institute of Botany, Chinese Academy of Sciences (HKAS), and Herbarium Mycology, Institute of Microbiology, Chinese Academy of Sciences (HMAS). The herbarium codes follow the Index Herbariorum.

### 2.2. Morphological Analysis

Basidiospores were examined in Melzer’s [61] reagent and measured in side view. At least 20 mature basidiospores were examined from basidiomata. Other microscopic structures were studied after these structures were soaked in 5% KOH and 1% Congo Red for 10 min. The ornamentation of the spores was observed under a scanning electron microscope (SEM, Coxem EM-30, Daejeon, South Korea). The structures were cut under a stereomicroscope (Leica S9E, Wetzlar, Germany), then observed and measured under a compound microscope (Leica DM 2500, Wetzlar, Germany). The measurements (and Q values) are given as (a) bec (d), in which “a” is the lowest value, “bec” covers a minimum of 90% of the values, and “d” is the biggest value. “Q” stands for the ratio of the length and width of a spore, and “Q ± av” represents the average Q of all spores ± sample standard deviation [61]. Other microscopic structures were treated in 5% KOH for 30 s and then observed in 1% Congo Red. Sections through the stipitipellis were taken from the middle of the stipe [64].

### 2.3. DNA Extraction, Amplification, and Sequencing

Dried specimens were used to extract genomic DNA using an EZgene^TM^ Fungal gDNA Kit (Biomiga, San Diego, CA, USA). Reaction mixtures (20 µL) contained 1 µL template DNA, 7 µL distilled water, and 1 µL (10 µM) of each primer and 10 μL 2 × Taq PCR StarMix with Loading Dye (Genstar, Kangrunchengye Biotech, Beijing, China). Three nuclear gene loci were amplified and sequenced: the universal primers ITS1 and ITS4 were used for amplification of the internal transcribed spacer (ITS) region of the ribosomal DNA, which includes spacer regions ITS1 and ITS2 and the ribosomal gene 5.8S; LROR and LR5 were the primers used for the amplification of LSU, which is a part of the ribosomal large subunit 28S region [65,66]; and RPB2-6F and RPB2-7CR were the primers used for amplification of the region between domains 6 and 7 of the second largest subunit of RNA polymerase II (rpb2) [65,66]. The PCR amplification reactions were performed on a T100 Thermal Cycler (T100™, Bio-Rad, Hercules, CA, USA). The ITS, LSU, and RPB2 regions were amplified by an initial denaturation step at 5 min at 95 °C, 35 cycles of 30 s at 94 °C, 30 s at 55 °C, 40 s at 55 °C, and a final extension stage of 5 min at 72 °C. PCR products were verified by 1% agarose gel electrophoresis and compared with 2 Kb DNA Markers [66]. The verified PCR products were purified and sequenced with the primers mentioned above at Sangon Biotech (Shanghai, China).

### 2.4. Sequence Alignment and Phylogenetic Analysis

The quality of the newly obtained sequences of three new specimens was checked manually by observing the chromatogram with BioEdit [67]. Three datasets (ITS, nrLSU, and RPB2) were generated from the representative (voucher) specimens of each species and used for phylogenetic analyses. Following preliminary analyses that placed the new species within *Lactifluus* subgenus *Lactifluus*, phylogenetic analyses were performed with the newly generated sequences and the sequences retrieved from GenBank [68], derived from the BLAST search (best match) of related *Lactifluus* species, complemented with other GenBank sequences of species of the sections within *Lactifluus* subgenus *Lactifluus* identified by De Crop [8] (Table 1). In this way, we selected 110 sequences of *Lactifluus* sect. *Lactifluus*, nine sequences of *Lactifluus* sect. *Tenuicystidiati*, three sequences of *Lactifluus* sect. *Allardii*, three sequences of *Lactifluus* sect. *Ambicystidiati*, nine sequences of *Lactifluus* sect. *Gerardii,* and six sequences of *Lactifluus* sect. *Piperati*, with *Auriscalpium vulgare*, *Bondarzewia montana,* and *Stereum hirsutum* being selected as outgroups [26].

### 2.5. Phylogenetic Analyses

All DNA datasets were aligned using the online version of MAFFT v.7 [69] (http://mafft.cbrc.jp/alignment/server/ accessed on 18 November, 2022) using the L-INS-i algorithm, then trimmed and edited in MEGA7.0 [70]. All phylogenetic analyses were performed in the PhyloSuite_v1.2.2 [71]. Phylogenetic analyses were conducted using the maximum likelihood (ML) strategy in IQ-TREE [72] and Bayesian inference (BI) in MrBayes v3.2.6 [73]. ML phylogenies were inferred using IQ-TREE under an edge-linked partition model for 5000 ultrafast [74] bootstraps, as well as the Shimodaira–Hasegawa–like approximate likelihood ratio test [75]. ModelFinder [76] was used to select the best-fit partition model (edge-linked) using the BIC criterion. The best-fit models were identified according to BI criteria (BIC): SYM + I + G4: ITS, K2P + I + G4: LSU, K2P + I + G4: RPB2. BI phylogenies were inferred using MrBayes 3.2.6 under a partition model (two parallel runs, 2,000,000 generations), in which the initial 25% of sampled data were discarded as burn-in. The phylogenies from ML and BI analyses were displayed using FigTree v1.4.3 [77].

## 3. Results

We generated 23 new sequences from the *Lactifluus* species studied, eight from each of the ITS and nLSU regions of rDNA and seven from the RPB2 region (Table 1). In the phylogenetic trees, ML and BI analyses produced highly similar topologies with comparable support values. The results inferred in the multilocus phylogeny (Figure 1) strongly supported the recognition of three new species, namely, *Lactifluus taibaiensis*, *Lactifluus jianbaensis*, and *Lactifluus qinggangtangensis*, based on phylogenetic studies with three regions (ITS, LSU, and RPB2).

### Taxonomy

***Lactifluus taibaiensis*** W.P. Zhang, A.M. Chen, and X.H. Xu, sp. nov., is shown in Figure 2. The MycoBank ID is 842968. The etymology refers to the collection site “Taibai”, and the holotype is HKAS 122860.

Pilei are 35.60–44.50 mm diameter and convex to planoconvex with a broadly depressed center. Velvet is mainly distributed on the edge, and orange (W3C) (#FFA500) is in the center when young, gradually becoming applanate to infundibuliform or concave; the surface is drying, smooth, dry, rugulose, velvety, and darker toward the center. The edge bends inward and is integral and brittle in consistency. Lamellae are decurrent, white (W3C) to cream (#FFFFCC), thick and brittle, and 2.00–3.40 mm broad; the edge is concolorous to marginate, furcate, and with different lengths. The attachment to the stipe varies from adnate to adnate with a decurrent tooth to decurrent and is milk white (#FEFCFF) after being bruised, with no discoloration reaction. The stipe is 46.50–81.10 × 9–14 mm, central, solid, dry, lighter colored than that of the pileus, rugulose, white at base, with a lot of white hyphae, cylindrical, and slightly curved; the latex is thick and milky white (#FEFCFF). The context is white (W3C) (#FFFFFF), and the taste is mild. Latex is abundant and sticky and changes from white to brown. Basidiospores are (2.85–)3.96–7.1(–8.6) × (3.19–)3.47–7.93(–8.28) μm, Q = (0.75–)0.83–1.23(–1.26), Q = 1.00 ± 0.13 μm, and they are subglobose and hyaline, with a strongly amyloid ornamentation composed of interconnected warts forming a complete reticulum up to 1.5 µm high (Figure 1). Basidia are 24.37–41.24 × 6.29–13.02 μm, Q = 2.29–2.99–3.98, with four sterigmata, which form four spores; and the sterigmata are 2.14–7.37 μm long. Pleuromacrocystidia are moderate to abundant, 51.07–63.83 × 4.44–7.89 µm, emergent up 30 µm, fusiform to subfusiform with fusoid, acuminate to subobtuse apices, originating from the subhymenial region. Pleuropseudocystidia are 1.68–4.11 µm wide. Cheilolamprocystidia are 31.53–58.66 × 6.37–6.04 µm, subcylindric to subfusiform with acuminate to subobtuse apices. Marginal cells are (15.35–)15.93–26.4(–29.03) × (1.64–)2.56–5.58(–5.76) μm, sublageniform, tortuous, tapering toward the apex, hyaline, fusoid, sometimes flexuous, thin-walled, and hyaline. Lactifers are 3.11–7.12 µm wide. The pileipellis is subcylindric to subfusiform to fusiform with rounded to acuminate apex; the margin is wavy; and the subpellis is pseudoparenchymatous, composed of rounded to elongated to somewhat irregularly shaped cells. The stipitipellis is composed of elements.

The known distribution is Taibai, Suiyang, Guizhou Province, China. The examined material is in China in Guizhou Province, Zunyi City, Suiyang (N 28°24′8″ E 107°5′31″, 1013.64 m), growing in groups on soil in association with *Castanopsis* spp., examined on 23 July 2020 by Xiuhong Xu (holotype is HKAS 122860, and isotype is HMAS 351908). (ITS = OL423562-OL423564, LSU = OL423575-OL423577, and RPB2 = OM030352-OM030354.)

***Lactifluus qinggangtangensis*** W.P. Zhang, A.M. Chen, and X.H. Xu, sp. nov., is shown in Figure 3. The MycoBank ID is 842971. The etymology refers to the collection site “Qinggangtang”, and the holotype is HKAS 122861.

Pilei are 24.12–52.73 mm diameter and are slightly concave in the center to convex to planoconvex with a broadly depressed center. Velvet is mainly distributed on the edge, and orange (W3C) (#FFA500) is in the center when young, gradually becoming applanate to infundibuliform or concave; the surface is smooth, dry, and velvety, with an uneven distribution. The edge bends inward and is integral and brittle in consistency. Lamellae are decurrent, white (W3C) to cream (#FFFFCC), thick and brittle, and dense; the edge is concolorous to marginate. The attachment to the stipe varies from adnate to adnate with a decurrent tooth to decurrent and is milk white (#FEFCFF) after bruising, with no discoloration reaction. The stipe is 39.12–59.09 × 9.41–11 mm, central, solid, dry, smooth, concolorous with the pileus, white at the base, cylindrical, and slightly curved, and the latex is thick and milky white (#FEFCFF). The context is white (W3C) (#FFFFFF), and the taste is mild. Basidiospores are (2.71–)3.24–8.45(–8.54) × (2.89–)3.04–8.11(–8.41) μm, Q = (0.86–)0.86–1.15(–1.33), Q = 1.05 ± 0.13 μm, and they are subglobose and hyaline, with a strongly amyloid ornamentation composed of interconnected warts forming a complete reticulum up to 1.42 µm high (Figure 2). Basidia are 22.88–44.9 × 5.83–11.00 μm, Q = 2.93–3.84–4.90, with four sterigmata; they form four spores; and sterigmata are 2.21–6.78 μm long. Pleuromacrocystidia are moderate to abundant, 48.89–90.84 × 7.63–17.79 µm, fusiform to subfusiform with fusoid, acuminate to subobtuse apices, originating from the sub-hymenial region. Pleuropseudocystidia are 0.94–5.27 µm wide. Cheilolamprocystidia are 31.14–57.75 × 4.46–9.79 µm and transparent. Marginal cells are 18.84–67.41 × 2.24–10.71 µm, sublageniform, tortuous, tapering toward the apex, hyaline, fusoid, sometimes flexuous, thin-walled, and hyaline. Lactifers are 3.23–9.50 µm broad. The pileipellis is broken hyphoepithelium to epithelium, often with round cells separated and scattered, forming a cutis between piles of round cells, rarely of globose cells, forming a continuous layer. The stipitipellis is a cutis of densely interwoven hyphae mostly parallel with the stipe length.

The known distribution is Qinggangtang, Suiyang, Guizhou Province, China. The examined material is in China, Guizhou Province, Zunyi City, Suiyang (N 28°20′50″ E 107°10′11″, 943.27 m), and is growing in groups on soil in association with *Castanopsis* spp., examined on 23 July 2020 by Xiuhong Xu (holotype is HKAS 122862, and isotype is HMAS 351909). (ITS = OL423568-OL423569, LSU = OL423581, OL655455, and RPB2 = OM030358.)

***Lactifluus jianbaensis*** W.P. Zhang, A.M. Chen, and X.H. Xu, sp. nov., is shown in Figure 4. The MycoBank ID is 842969. The etymology refers to the collection site “Jianba”, and the holotype is HKAS 122862.

Pilei are 42.81–46.25 mm diameter, and are slightly concave in the center to convex to planoconvex with a broadly depressed center, dark orange (W3C) in the center, mango orange (#FF8040) on the edge, and velvet in the center when young, gradually becoming applanate to infundibuliform or concave. The surface is dry and smooth. The edge bends flat and is brittle in consistency. Lamellae are decurrent, white (W3C) to cream (#FFFFCC), thick and brittle, and dense, and the edge is concolorous to marginate. The attachment to the stipe varies from adnate to adnate with a decurrent tooth to decurrent, and the milk is colorless to white (#FEFCFF) and turns brown in a few minutes after being bruised. The stipe is 56.25–60.94 × 8.59–13.75 mm, central, solid, dry, smooth, dark orange (W3C), uneven in color, white at the base, cylindrical, and slightly curved. The latex is abundant and watery. Basidiospores are (5.09–)5.25–7.52(–7.68) × (2.85–)3.75–7.24(–8.00) μm, Q = (0.80–)0.86–1.40(–1.79), Q = 1.06 ± 0.13 μm, subglobose, and hyaline, with a strongly amyloid ornamentation composed of interconnected warts forming a complete reticulum up to 2.17 µm high. Basidia are 31.32–44.77 × 11.59–16.06 μm, with four sterigmata, and form four spores. Pleuromacrocystidia are moderate to abundant, 65.38–102.98 × 5.19–11.67 µm, fusiform to subfusiform with fusoid, acuminate to subobtuse apices, originating from the sub-hymenial region. Marginal cells are 18.13–28.24 × 5.19–11.67 µm, sublageniform, tortuous, tapering toward the apex, hyaline, fusoid, sometimes flexuous, thin-walled, and hyaline. Lactifers are 2.55–6.77 µm broad. The pileipellis is subcylindric to subfusiform to fusiform with rounded to acuminate apex; the margin is wavy, composed of rounded to elongated to somewhat irregularly shaped cells. The stipitipellis is composed of elements.

The known distribution is Jianba, Suiyang, Guizhou Province, China. The examined material is in China, Guizhou Province, Zunyi City, Suiyang (N 29°0′24″ E 107°43′50″, 1044.54 m), growing in groups on soil in association with *Pinus* sp., examined on 12 October 2020 by Xiuhong Xu (holotype is HKAS 122862, and isotype is HMAS 351910). (ITS = OL423565-OL423567, LSU = OL423578-OL423580, RPB2 = OM030355-OM030355.)

## 4. Discussion

In this study, three new accessions are identified as novel species of *Lactifluus* sect. *Lactifluus* in terms of both morphology and phylogeny. *Lactifluus taibaiensis*, with its sister species *Lactifluus rugiformis* from South Korea [27]; *Lactifluus jianbaensis*, with its sister species *Lactifluus acicularis* from Thailand [15]; and *Lactifluus qinggangtangensis*, with its sister species *Lactifluus pinguis* from Thailand [15], form well-separated clades in the resultant phylogram, which indicate the distinct phylogenetic positions of the three new species in *Lactifluus* sect. *Lactifluus.*

*Lf. taibaiensis* is an orange milkcap, with a rugulose stipe, similar to its sister species *Lf. rugiformis* [27]. There are also many other characteristics to distinguish one another, with *Lf*. *taibaiensis* being lighter in the color of the pileus (orange (W3C) (#FFA500) vs. rusty orange (6C8–7C8)) and having a higher ratio of stipe length/pileus diameter (1.3–1.8 vs. 0.7) and different colors of lamellae (cream vs. cream to pale orange). When comparing micromorphologic features between *Lf. taibaiensis* and *Lf. rugiformis*, the basidiospores of the former are more subglobose (0.75–1.26 vs. 1.01–1.09) (Table 2). *Lf. jianbaensis* differs from its sister species *Lf. acicularis* in terms of the pileus color (dark orange (W3C) vs. brown (6D5)), the diameter of the pileus (43–46 mm vs. 35–85 mm), and the height of the spore ornamentation of subglobose basidiospores (2.17 μm vs. 1.40 μm); this can also be distinguished from *Lf. longipilus.* The main difference between *Lf. qinggangtangensis* and *Lf. pinguis* is the smaller diameter of the pileus of the former (24.12–52.73 mm vs. 40–80 mm), smaller basidiospores (2.71–8.54 × 2.89–8.41 μm vs. 8.0–9.0–9.1–10.2(–10.5) × 7.4–8.3–8.4–9.4(–9.6) μm), smaller ridges (1.42 µm vs. 2.0 µm), and smaller basidia (22.88–44.9 × 5.83–11.00 μm vs. 40–65 × 11–14 μm). *Lf. jianbaensis* can be distinguished from *Lf. acicularis* by the height of the spore ornamentation, which can be up to 2.17 µm in *Lf. jianbaensis* but only up to 1.1–1.4 μm in *Lf. acicularis.*

It is noteworthy that three new *Lactifluus* species were described in Guizhou, southwest China. As a result of the investigations into *Lactifluus* resources in Guizhou over the years, Guizhou was found to be particularly rich in *Lactifluus* spp. (Table 2), namely, *Lactifluus leoninus* Verbeken and E. Horak Verbeken, in Verbeken, Nuytinck, and Buyck [78], *Lactifluus bhandaryi* Verbeken and De Crop, *Lactifluus subpiperatus* (Hongo) Verbeken [79], *Lactifluus pseudoluteopus* X.H. Wang and Verbeken, X.H. Wang [80], and *Lactifluus volemus* [79,81,82,83]. Many *Lactifluus* are considered to be edible mushrooms and are sold at the local markets and along roadsides, fresh, dried, or boiled. In addition to *Lactifluus*, its related milkcap genus *Lactarius* is also very rich in subtropical China, and several new species were described recently [84,85,86]. So, the diversity of ectomycorrhizal milkcap mushrooms is rich these areas.

## Figures and Tables

**Figure 1 jof-09-00122-f001:**
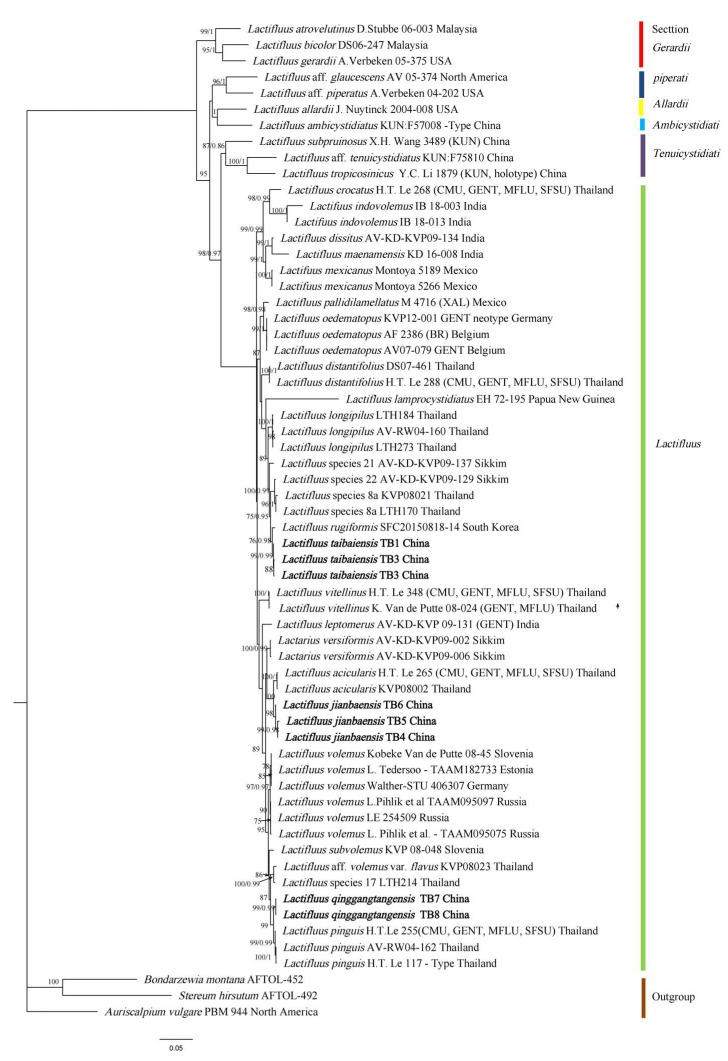
Phylogram for *Lactifluus* sect. *Lactifluus* generated from maximum likelihood analysis of ITS, LSU, and RPB2 sequence data. Bootstrap support values for maximum likelihood and maximum parsimony greater than 50% and posterior probabilities from Bayesian inference ≥0.95 are given above the branches. The new species are presented in bold type.

**Figure 2 jof-09-00122-f002:**
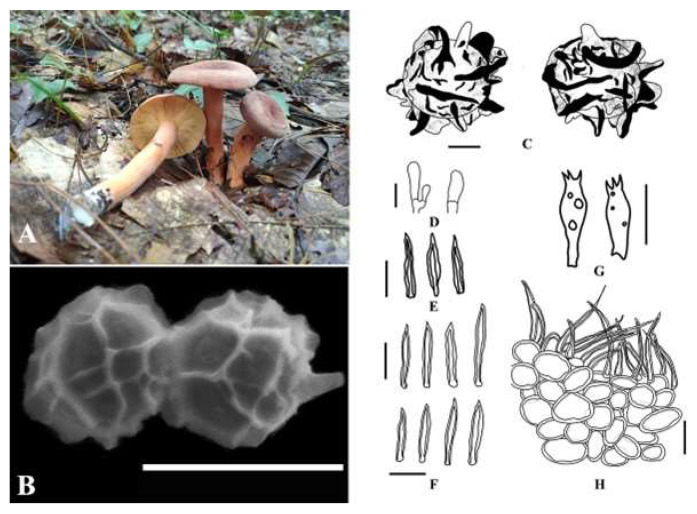
*Lactifluus taibaiensis* microscopic characteristics: (**A**) Fresh basidiomata (holotype). (**B**) SEM microphotographs. (**C**) Basidiospores. (**D**) Marginal cells. (**E)** Pleurocystidia. (**F**) Cheilocystidia. (**G**) Basidia. (**H**) Pileipellis. Scale bars: 4 µm (**B**), 3 µm (**C**), and 30 µm (**D**–**H**).

**Figure 3 jof-09-00122-f003:**
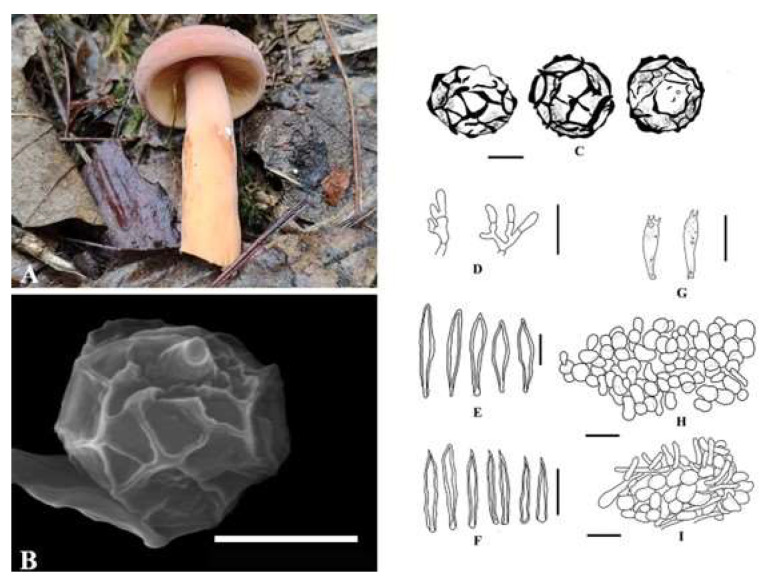
*Lactifluus qinggangtangensis* microscopic characteristics: (**A**) Fresh basidiomata (holotype). (**B**) SEM microphotographs. (**C**) Basidiospores. (**D**) Marginal cells. (**E**) Pleurocystidia. (**F**) Cheilocystidia. (**G**) Basidia. (**H**) Stipitipellis. (**I**) Pileipellis. Scale bars: 4 µm (**B**), 3 µm (**C**), and 30 µm (**D**–**I**).

**Figure 4 jof-09-00122-f004:**
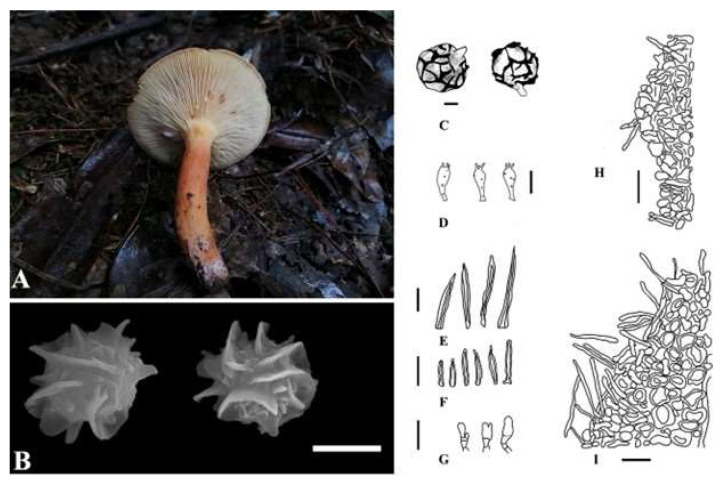
*Lactifluus jianbaensis* microscopic characteristics: (**A**) Fresh basidiomata (holotype). (**B**) SEM microphotographs. (**C**) Basidiospores. (**D**) Basidia. (**E**) Pleurocystidia. (**F**) Cheilocystidia. (**G**) Marginal cells. (**H**) Stipitipellis. (**I**) Pileipellis. Scale bars: 4 µm (**B**), 3 µm (**C**), and 30 µm (**D**–**I**).

**Table 1 jof-09-00122-t001:** Specimen and GenBank accession numbers of DNA sequences used in the molecular analyses. The arrangement of the subgenera and sections in the table follows their position in the concatenated phylogeny of the genus *Lactifluus* subgenus *Lactifluus* (Figure 1).

Species	Voucher Specimen No.	Locality	ITS	LSU	RPB2
*Lactifluus* subgenus *Lactifluus*					
*Lactifluus* sect. *Lactifluus*					
*Lf. acicularis*	H.T. Le 265 (CMU, GENT, MFLU, SFSU)	Thailand	HQ318277	HQ318196	HQ328926
*Lf. acicularis*	KVP08002	Thailand	HQ318226	HQ318132	HQ328869
*Lf.* aff. *tenuicystidiatus*	KUN:F75810	China	KC154105	KC154131	KC154157
*Lf.* aff. *tenuicystidiatus*	KUN:F75810	China	KC154105	KC154131	KC154157
*Lf.* aff. *volemus* var. *flavus*	KVP08023	Thailand	HQ318227	HQ318133	HQ328870
*Lf. crocatus*	H.T. Le 268 (CMU, GENT, MFLU, SFSU)	Thailand	HQ318266	HQ318181	HQ328917
*Lf. dissitus*	AV-KD-KVP09-134	India	JN388978	JN389026	JN375628
*Lf. distantifolius*	H.T. Le 288 (CMU, GENT, MFLU, SFSU)	Thailand	HQ318274	HQ318193	
*Lf. distantifolius*	DS07-461	Thailand	HQ318223	HQ318124	HQ328866
*Lf. indovolemus*	IB 18-013	India	MN005117		
*Lf. indovolemus*	IB 18-003	India	MN005115		
*Lf. jianbaensis*	TB 4	China	OL423565	OL423578	OM030355
*Lf. jianbaensis*	TB 5	China	OL423566	OL423579	OM030356
*Lf. jianbaensis*	TB 6	China	OL423567	OL423580	OM030357
*Lf. lamprocystidiatus*	EH 72-195	Papua New Guinea	KR364015		
*Lf. leptomerus*	AV-KD-KVP 09-131 (GENT)	India	JN388972	JN389023	JN375625
*Lf. longipilus*	LTH273	Thailand	HQ318276	HQ318195	HQ328925
*Lf. longipilus*	LTH184	Thailand	HQ318256	HQ318169	HQ328905
*Lf. longipilus*	AV-RW04-160	Thailand	HQ318235	HQ318143	HQ328880
*Lf. maenamensis*	KD 16-008	India	MF928075		
*Lf. mexicanus*	Montoya 5189	Mexico	MK211179	MK211188	MK258869
*Lf. mexicanus*	Montoya 5266	Mexico	MK211180	MK211189	MK258870
*Lf. oedematopus*	KVP12-001 GENT neotype	Germany	KJ210065	KJ210066	KJ210068
*Lf. oedematopus*	AF 2386 (BR)	Belgium	JQ753876	JQ348324	JQ348260
*Lf. oedematopus*	AV07-079 GENT	Belgium	JQ753835	JQ348270	JQ348131
*Lf. pallidilamellatus*	M 4716 (XAL)	Mexico	JQ753824	JQ348268	
*Lf. qinggangtangensis*	TB 7	China	OL423568	OL423581	OM030358
*Lf. qinggangtangensis*	TB 8	China	OL423569	OL655455	
*Lf. pinguis*	AV-RW04-162	Thailand	HQ318221	HQ318121	HQ328863
*Lf. pinguis*	H.T. Le 117—Type	Thailand	HQ318211	HQ318111	HQ328858
*Lf. pinguis*	H.T. Le 255 (CMU, GENT, MFLU, SFSU)	Thailand	HQ318263	HQ318178	HQ328914
*Lf. rugiformis*	SFC20150818-14	South Korea	MN215387	MN215343	MN212835
*Lf.* sect. *Tenuicystidiati*					
*Lf*. species 17	LTH214	Thailand	HQ318249	HQ318158	HQ328894
*Lf.* species 21	AV-KD-KVP09-137	Sikkim	JN388958	JN389027	JN375629
*Lf.* species 22	AV-KD-KVP09-129	Sikkim	JN388957	JN389021	JN375623
*Lf.* species 8a	KVP08021	Thailand	HQ318233	HQ318140	HQ328877
*Lf.* species 8a	LTH170	Thailand	HQ318252	HQ318165	HQ328902
*Lf. subpruinosus*	X.H. Wang 3489 (KUN)	China	KC154110	KC154136	KC154162
*Lf. subvolemus*	KVP 08-048	Slovenia	JQ753927	JQ348379	JQ348241
*Lf. taibaiensis*	TB 1	China	OL423562	OL423575	OM030352
*Lf. taibaiensis*	TB 2	China	OL423563	OL423576	OM030353
*Lf. taibaiensis*	TB 3	China	OL423564	OL423577	OM030354
*Lf. versiformis*	AV-KD-KVP09-002	Sikkim	JN388966	JN389030	JN375631
*Lf. versiformis*	AV-KD-KVP09-006	Sikkim	JN388965	JN389033	JN375633
*Lf. vitellinus*	H.T. Le 348 (CMU, GENT, MFLU, SFSU)	Thailand	HQ318251	HQ318164	HQ328900
*Lf. vitellinus*	K. Van de Putte 08-024 (GENT, MFLU)	Thailand	HQ318236	HQ318144	HQ328881
*Lf. volemus*	L. Pihlik et al.—TAAM095075	Russia	JQ753905	JQ348357	JQ348219
*Lf. volemus*	Walther—STU 406307	Germany	JQ753909	JQ348361	JQ348223
*Lf. volemus*	L. Tedersoo—TAAM182733	Estonia	JQ753907	JQ348359	JQ348221
*Lf. volemus*	LE 254509	Russia	JQ753937	JQ348388	
*Lf. volemus*	L. Pihlik et al.—TAAM095097	Russia	JQ753906	JQ348358	JQ348220
*Lf. volemus*	Kobeke Van de Putte 08-45	Slovenia	JQ753953		
*Lf.* sect. *Allardii*					
*Lf. allardii*	J. Nuytinck 2004-008	USA	KF220016	KF220125	KF220217
*Lf.* sect. *Ambicystidiati*					
*Lf. ambicystidiatus*	KUN:F57008—Type	China	NR_155311	NG_060287	KC154148
*Lf.* sect. *Gerardii*					
*Lf. atrovelutinus*	D. Stubbe 06-003	Malaysia	GU258231	GU265588	GU258325
*Lf. bicolor*	DS06-247	Malaysia	JN388955	JN388987	JN375590
*Lf. gerardii*	A.Verbeken 05-375	USA	GU258254	GU265616	GU258353
*Lf.* sect. *Piperati*					
*Lf.* aff. *glaucescens*	AV 05-374	North America	KF220049	KF220150	KF220236
*Lf.* aff. *piperatus*	A.Verbeken 04-202	USA	KF220021	KF220127	KF220220
Outgroup					
*Auriscalpium vulgare*	PBM_944	North America	DQ911613	DQ911614	AY218472
*Bondarzewia montana*	AFTOL_452	No data	DQ200923	DQ234539	AY218474
*Stereum hirsutum*	AFTOL_492	No data	AY854063	AF393078	AY218520

**Table 2 jof-09-00122-t002:** Synopsis of sister species to the new *Lactifluus* species reported here with respect to distribution and morphological features.

Species	*Lf. rugiformis*	*Lf. pinguis*	*Lf. acicularis*
Location	Korea	Thailand	Thailand
Pileus length (mm)	50–110	35–85	3.3–4.6
Pileus color	Rusty orange (6C8–7C8) tinged with a more brownish color	Yellowish-orange-brown (5C7–5C8), brown (6D5)	Yellowish-orange-brown (5C7–5C8), brown (6D5)
Lamella breadth (mm)	Three broad, rarely furcate, with numerous lamellula of different length	2–4	Narrow to rather broad (1.5– 6 mm)
Lamella color	Cream to pale orange	Whitish to cream	Cream (4A3–4A4), discoloring to brown (6D5– 6E5) to grayish-brown (5C3–5C4) when damaged
Stipe (mm)	30–70 × 15–20	40–95 × 10–15	45–85 × 5–15
Stipe color	Concolorous with pileus	Concolorous with pileus	Yellowish-orange (4A5–5A5),brownish-orange (6C8–6D8) to grayish-brownish-orange (6C5–5C6–6B5–6B6–6C6), or brown (6D4–6D5)
Latex	Abundant, sticky, white turning dark brown	Copious, sticky, white, unchanging when isolated	White
Basidiospores (μm)	7.1–8.4–9.6 × 6.7–7.9–9.2, Q = 1.01–1.05–1.09, globose to subglobose	8.0–9.0–9.1–10.2(–10.5) × 7.4–8.3–8.4– 9.4(–9.6)	7.0–7.9–8.5–9.1(–9.3) × 6.5–7.2–7.8– 8.5, subglobose (Q = 1.01–1.08–1.10–1.21)
Basidia (μm)	49.5–60 × 9–12.5	40–65 × 11–14	40–60 × 9–12
Pileipellis (μm)	20–68 × 2.5–4.0, cell wall 0.5–1.5, thick, erect	50–140	50–120, thick
References	[21]	[15]	[15]

## Data Availability

Not applicable.

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
