# Peer review of "Three New Species of Lactifluus (Basidiomycota, Russulaceae) from Guizhou Province, Southwest China"

_jof, 2023, doi:10.3390/jof9010122_

Round 1
Reviewer 1 Report
Some comments on the text see also attached file
Line 19 – as seems to be in bold
Line 21 – “white latex on lamellae”, this is not strictly the case I suppose it is latex from the damaged lamellae.
Line 24 – through field studies?
Line 26 – Is it nuclear or ribosomal as well?
Line 27 to 32 - Using "lighter, more subglobose, smaller" are comparative but there are no taxa mentioned to compare. Lighter than what, more subglobose than what, smaller than what?
Line 49 – Jumping to the subordinal level here is difficult to follow without reference to the order since all previous mention has been of the genus. This needs to be put into context
Line 61 – These species presumably referring to Lactifluus.
Line 65 – edible resource
Line 66 to 67 – here there is a jump. Having never mentioned edibility you are now saying these are edible except those with unpleasant tastes that may cause distress.
Line 68 – more common that what?
Line 78 and following - Why all this when it is just Guizhou Province where these were collected. I recomment scaling this back.
Line 104 - Other microscpic structures were studied after portions of the spormome were soaked in…
Line 148 – New sequences are indicated in boldface type?
Line 165 – remove built
Line 192 – concolorous, marginate
Line 197 – taste?
Line 251 – Holotype should be indicated here too. Since these are parts of the same single collection.
Line 314 - this is an odd construction perhaps “Synopsis of species sister to…”
Line 316 – Why is it noteworthy that these three species were discovered? Is it because of the previous work by Verbeken et al. The authors are adding to the high diversity that has already been noted. Are these species edible?
Given the nature of the paper the references cited are extensive and in some cases not apparent why they are included.
There might be more mention of edibility. There is reference throughout to these fungi as a resource. Perhaps some aspects of the commerce involved.
One of the recent recommendations for describing species is to have three collections or at least more than one. All of these are based on single collections.

Reviewer 2 Report
With reference to email received from said journal, you have asked me to review “Three New Species of Lactifluus (Basidiomycota, Russulaceae) from Guizhou Province, Southwest Chin”. This paper highlighted the extensive fungal diversity that is yet being explored in China and report specimens for the first time that focuses on their taxonomical based classification. It is necessary that modern taxonomic methods and databases are maintained to carry out further research on such important organisms, which are mostly neglected in Asian region. The present work is contributing to this aspect by identifying the said fungal species by combination of Evolutionary and Cladistic systematic. Since last four decades, Phylogenetic Systematics has been replacing traditional systematic. The present work is a good illustration of how each approach could feed the other.
The body/text is well presented with numerous references to previous works. The layout of the results, photographs and drawings are excellent; discussion part include relevant literature. To conclude my review, I am impressed with paper and is worth being published. I find this work as original and demanded.
Author Response
Thank you to the reviewers for comments on our articles, and we are honored to have your recognition of our work.